# Artificial Intelligence Applications to Improve the Treatment of Locally Advanced Non-Small Cell Lung Cancers

**DOI:** 10.3390/cancers13102382

**Published:** 2021-05-14

**Authors:** Andrew Hope, Maikel Verduin, Thomas J Dilling, Ananya Choudhury, Rianne Fijten, Leonard Wee, Hugo JWL Aerts, Issam El Naqa, Ross Mitchell, Marc Vooijs, Andre Dekker, Dirk de Ruysscher, Alberto Traverso

**Affiliations:** 1Department of Radiation Oncology, University of Toronto, Toronto, ON 5MT 1P5, Canada; andrew.hope@rmp.uhn.ca; 2Radiation Medicine Program, Princess Margaret Cancer Centre, University Health Network, Toronto, ON 5MT 1P5, Canada; 3Department of Radiation Oncology (Maastro) GROW School for Oncology and Developmental Biology, Maastricht University Medical Centre+, 6229 ET Maastricht, The Netherlands; m.verduin@maastrichtuniversity.nl (M.V.); ananya.choudhury@maastro.nl (A.C.); rianne.fijten@maastro.nl (R.F.); leonard.wee@maastro.nl (L.W.); marc.vooijs@maastrichtuniversity.nl (M.V.); andre.dekker@maastro.nl (A.D.); dirk.deruysscher@maastro.nl (D.d.R.); 4Department of Radiation Oncology, H. Lee Moffitt Cancer Center and Research Institute, Tampa, FL 33612, USA; Thomas.Dilling@moffitt.org; 5Artificial Intelligence in Medicine (AIM) Program, Mass General Brigham, Harvard Medical School, Boston, MA 02115, USA; haerts@bwh.harvard.edu; 6Department of Radiation Oncology, Brigham and Women’s Hospital, Dana-Farber Cancer Institute, Harvard Medical School, Boston, MA 02115, USA; 7Radiology and Nuclear Medicine, CARIM & GROW, Maastricht University, 6228 ET Maastricht, The Netherlands; 8Department of Machine Learning, H. Lee Moffitt Cancer Center and Research Institute, Tampa, FL 33612, USA; Issam.ElNaqa@moffitt.org (I.E.N.); Ross.Mitchell@moffitt.org (R.M.)

**Keywords:** lung cancers, artificial intelligence, radiomics, deep learning, clinical decision aids

## Abstract

**Simple Summary:**

The management of locally advanced (stages II–III) non-small cell lung cancer patients is very challenging because of poor survival rates and patient/tumor heterogeneity. In this review, we identify the critical points that can be addressed by artificial intelligence (AI) algorithms to improve care of these patients and to present a roadmap for AI applications that will support better treatments.

**Abstract:**

Locally advanced non-small cell lung cancer patients represent around one third of newly diagnosed lung cancer patients. There remains a large unmet need to find treatment strategies that can improve the survival of these patients while minimizing therapeutical side effects. Increasing the availability of patients’ data (imaging, electronic health records, patients’ reported outcomes, and genomics) will enable the application of AI algorithms to improve therapy selections. In this review, we discuss how artificial intelligence (AI) can be integral to improving clinical decision support systems. To realize this, a roadmap for AI must be defined. We define six milestones involving a broad spectrum of stakeholders, from physicians to patients, that we feel are necessary for an optimal transition of AI into the clinic.

## 1. Introduction

Stage II–III (locally advanced) non-small cell lung cancer (LA-NSCLC) represents around one third of newly diagnosed lung cancer patients [1]. Management of these patients is challenging due to poor survival rates and inter-patient heterogeneity. These patients have widely variable presentations, ranging from resectable tumors with microscopic lymph node metastases to bulky, unresectable disease. The pre-existing cardiopulmonary comorbidities observed in most of these patients can also complicate treatment decision-making [2]. New treatments are urgently needed to improve the overall survival (OS) by reducing both local tumor recurrence and the development of distant metastases while balancing the risk of life-threatening treatment-induced side effects, such as infections and cardiopulmonary toxicity. Ideally, treatments should be tailored to the unique biological properties of the patient’s tumor, referred to as “personalized medicine” [3]. Finally, the treatment strategy should not be conceived as a single timepoint decision but modified over time, always questioning if the treatment delivered today still provides benefits while minimizing added toxicities.

Treatment complexity has increased in the last decade as new strategies such as immunotherapy, molecularly targeted therapy, or advanced radiotherapy (RT) techniques (e.g., proton therapy) have been introduced into clinical practice or are under evaluation [4].

In unresectable LA-NSCLC patients, concurrent chemo-RT remains the gold standard treatment. However, the results are far from optimal: the progression-free survival (PFS) is about 15 months and the five year OS is about 25–35% [5]. This aggressive form of treatment is associated with acute toxicities, such as grade 3–4 esophagitis, radiation pneumonitis, and life-threatening infections. Therefore, younger patients with minimal or no comorbidities at presentation benefit the most from this strategy. Unfortunately, only a small percentage of LA-NSCLC patients (only 41% in a Dutch population study [6]) are eligible for concurrent RT [7]. Consolidation approaches with the addition of systemic therapy (e.g., docetaxel), radiation dose escalation, and targeted agents have not improved the OS [8,9]. However, the addition of one year of durvalumab to chemotherapy improves the four year OS significantly by about 15% while maintaining quality of life [10]. Other radiotherapeutic strategies, such as alternative fractionation schedules or proton therapy are also under evaluation. Nevertheless, introducing adjuvant or consolidation treatment options only brings benefits to patients at high risk of developing disease relapse but could lead to unnecessary toxicities for patients at lower risk as well as a decrease in cost effectiveness.

Despite the significant improvement in OS with the addition of durvalumab, there is still room for improvement in ultimate cure rates. Some of these patients do ultimately fail treatment—might they benefit from some other therapy? Discovering biomarkers to allow for an accurate stratification of these patients is still an unmet need.

Just as the advent of computed tomography scans produced a “data explosion” [11], we could say that recent advances in cancer diagnostics and therapy have produced a wealth of imaging and molecular data that require filtering to determine optimal medical treatment for any given patient. Rather than overly simplifying to focus on a limited number of data elements (such as tumor stage or patient performance status) to optimize treatment decision-making, patients should be evaluated as “sources of big data.” These data derive from multiple sources, such as electronic health records, patient-reported outcomes, laboratory tests, and medical images.

Medical images are routinely used in radiation oncology for diagnosis, treatment planning, treatment delivery, and disease follow up. Image-guided RT (IGRT) is a method of radiation therapy that incorporates imaging techniques prior to (or potentially during) each treatment session to ensure accurate treatment setup and delivery. Image-guided adaptive radiotherapy (IGART) is related to the concept of IGRT. In IGART, scans acquired before the delivery of a treatment on the linear accelerator (linac) are used to verify the alignment of the patient prior to delivery in response to anatomical variations observed during the therapy process. Furthermore, IGART allows for real-time revision of the treatment plan if the dose to an at-risk organ exceeds a predetermined threshold. Examples of the above include inter- and intra-treatment variations of patient/organ shapes and positions caused by patient organ physiological motion and deformation (e.g., gastric filling, colon peristalsis, etc.).

While IGRT and IGART are powerful tools to guarantee treatment delivery under conditions, mimicking the ones existing at the timepoint of treatment planning, IGRT and IGART promise to deliver on yet another unfulfilled need. These tools allow us to rethink the role of medical images as more than instruments for visual inspection or for accomplishing daily clinical tasks: these images are sources of highly mineable data that can reveal unique patient biomarkers. Images represent a dynamic array of unique data, not only about the tumor itself but also about the surrounding tissues and the relationship among the tumor and organs, longitudinally, across time. Being able to extract image-derived biomarkers that can quantify the evolution of tumors and anatomy in patients during treatment, referred to as “radiomics and deep radiomics” [12], can support quantification of the effectiveness of the treatment and allows us to revise our treatment strategy, if necessary. Biomarkers need to be extracted from both the target volumes and from normal tissues inside the radiation field: the organs at risk (OAR). Correlations among image-derived biomarkers, other clinical prognostic factors, biological properties of the tumor, and evidence arising from previously published studies should be amalgamated to support future causality studies (randomized clinical trials). In the unmet clinical need presented earlier, understanding the biology of both responders and non-responders to first-line therapy is of primary importance for our understanding and trust in such biomarkers.

Artificial intelligence (AI) can be a powerful tool for interpreting and guiding medical diagnoses, treatment, and follow-up. For LA-NSCLC patients, we envision using a combination of AI solutions for optimization of the IGRT workflow. It is important to emphasize that we do not see AI as a “virtual clinician” but rather as a supportive tool to free clinicians from time-consuming tasks in the RT workflow and to augment their decision-making capabilities.

In this paper, we present how AI can (A) be used to support a more advanced adaptive RT concept while remaining human- and patient-centered; (B) improve clinical practice and the current RT workflow by introducing automation for time-consuming tasks; and (C) bridge multiple sources of data, with a specific focus on biomarkers derived from imaging, clinical factors, and tumor biology. We also discuss the challenges that AI faces when dealing with medical images acquired from routine care, with a dedicated focus on the impact of the lower quality of in-treatment room imaging on the robustness of biomarker identification. As a case in point, we incorporate the above applications to the management of unresectable LA-NSCLC.

The outline of this paper is depicted in Figure 1. In the first section, we focus on the role of imaging, adaptive RT, and biomarkers as an unmet clinical need in the management of LA-NSCLC (critical point 1). In the second section, we present AI technologies that improve image quality and how this supports the extraction of more robust biomarkers (critical point 2). In the third section, we present the most recent AI-driven technologies to extract biomarkers from medical images as well as to link these biomarkers to tumor biology (critical point 3). In the fourth section, we describe how to incorporate these methodologies into clinical practice to improve and optimize the current IGRT workflow (critical point 4). In the last section, we focus on requirements for smoother implementation of the above technologies as decision support systems in the clinic (critical point 5). For all of the above points, we require the interaction among multiple stakeholders: physicians, patients, researchers, radiation therapists, and medical physicists.

## 2. Unmet Clinical Needs in the Management of LA-NSCLC Patients: Role of Imaging, Adaptive RT, and Biomarkers

There are a number of unmet clinical needs that fall into five categories: Section 2.1 presents the general automation of clinical processes; Section 2.2 presents improved prognostication regarding expected patient outcomes in the absence of recurrent disease; Section 2.3 presents the characterization/prediction of a malignant disease course; Section 2.4 presents the characterization/prediction of treatment toxicity; and Section 2.5 presents an integration of all of these predictive/prognostic metrics into a comprehensive “personalized” prediction.

### 2.1. General Automation of Clinical Processes

While other oncologic specialties such as surgery and medical oncology rely on images as part of their feedback process in preparing for treatment (surgery) or evaluating treatment response (post-surgery and chemotherapy), their actual treatments are physically planned outside the imaging space (operating room and chemotherapy suite). Radiation oncologists live in the imaging space during treatment planning and make many treatment decisions in that digital domain. This creates a unique “digital medical environment” in which AI/machine learning (ML) can have nearly full access to the same information as the clinician (minus the physical examination). As a result, AI has the potential to be highly integrated into standard radiation oncology processes. There are multiple areas where AI may be immediately useful in the “general” category:Contouring. As part of radiation treatment planning, contouring is a physician-directed image classification, whereby tumor targets (gross tumor volume (GTV) and clinical target volume (CTV)) are manually segmented as discrete and distinct from OAR or normal anatomy. Unfortunately, this is also a time-consuming process. AI has been shown to be capable of image classification in the clinical space [12,13,14,15], and rapid segmentation via AI represents a potential force-multiplier to enable individual radiation oncologists to evaluate and treat more patients per capita. This image-processing task has long been recognized as an important unmet clinical need, which has been a research target for numerous groups. Our work focused on the introduction in clinical practice of auto-contouring for RT OAR, comparing both atlas-based and deep learning algorithms [16], as well as on the automation of RT target volume delineations [17]. Recent results showed that deep learning algorithms can outperform expert technicians on lesion segmentation tasks [18].Automated review/classification of clinical records. Interestingly, as the digital domain for electronic medical records becomes the standard, the number and diverse types of clinical records accessible to AI for each patient is growing. However, the data in these resources are often unstructured (not discrete). As a result, there may be a role for natural language processing (NLP) in the review of clinical records, for example, to automatically extract comorbid illness, to identify pathologic diagnosis and/or biomarkers of relevance (epidermal growth factor receptor (EGFR), K-RAS, anaplastic lymphoma kinase (ALK), and programmed death ligand (PDL1)) from pathologic data, or to identify previous radiotherapy treatments that may identify risks for retreatment. We anticipate that the automated classification of free text medication records into a structured format will become increasingly important to monitor interactions between treatments and outcomes. For example, we have investigated the role of NLP in homogenizing radiological reports and in extracting standardized knowledge, such as the automated classification of tumor T stage from free text [19]. This work was extended to classify lesions as well as other characteristics from lung radiological reports. Despite promising results, the difficulties encountered during these studies underline the need for standardized nomenclature in medical records by the use of dedicated ontologies and semantic web techniques. This has been acknowledged by the European Society for Therapeutic Radiation Oncology (ESTRO) [20,21], the American Association of Physics in Medicine (AAPM) [22], and the American Society for Radiation Oncology (ASTRO) and resulted in the formation of working groups to establish these guidelines. We have also developed a deep learning system to automatically identify and extract tumor site and histology from free-text pathology reports. Our system predicts ICD-O-3 codes and preferred phrases with accuracies comparable to human experts [23]. In LA-NSCLC patients, it identifies lung subregions and tumor subtypes.Standardized data collection/ontological classification. There is also an interesting interaction between the automation described above and the subsequent usability of the data obtained. Automation not only leads to efficiency gains but also, generally, leads to more standardized/ontological data collection, which in turn may lead to better prognostication and prediction. As an example, a recent paper using AI-based automated heart segmentation led to a better prediction of dose-related cardiac toxicity in a pivotal trial on advanced lung cancer patients (RTOG 0617) compared to human heart segmentations, likely due to interobserver variation [24].

### 2.2. Improved Prognostication Regarding Expected Patient Outcomes in the Absence of Recurrent Disease

An assessment of patient life expectancy can strongly influence the choice of curative (or noncurative) treatments for each patient. Classically, if a patient has a limited life expectancy from a comorbid illness (advanced dementia, Parkinson’s disease, cardiopulmonary disease, and competing malignancies), it would be logical to consider this when evaluating whether to attempt a highly toxic treatment for a patient with a newly diagnosed LA-NSCLC. Frailty assessments and life-expectancy evaluations have long been largely “clinical” with limited reproducibility [25]. AI efforts to link all available medical record information in conjunction with objective imaging findings or radiomics parameters (i.e., muscle wasting or cachexia) may be a fruitful area of research. Already, cachexia as a biomarker of frailty has been shown to be well correlated with outcomes in a number of diseases and may be used to establish a “baseline survival estimate” for an LA-NSCLC patient being considered for curative chemoradiation therapy [26,27].

### 2.3. Characterization/Prediction of Malignant Disease Course: Disease Response to Treatments

Regardless of the underlying patient condition, some lung cancers are more biologically aggressive than others. Predicting the intrinsic “aggressiveness of disease” from a biologic standpoint would be of great interest to most clinicians as it would again allow clinicians to discuss the risks/benefits of treatment in a more personalized way. AI methods to integrate tumor growth rates, risks of distant metastases, disease resistance to specific treatments (e.g., chemotherapy or RT resistance), or ways to better predict the natural history of disease may be crucial. Deconvolving these natural history markers from the influence of treatment will require huge amounts of data and AI/ML methods to manage the complexity of those combined data sets. Preliminary work has been conducted, for example, to explore correlations between imaging features and tumor phenotype. Examples of this research in lung cancers found associations among multiple types of radiomic features extracted from the primary tumors and patients’ prognoses [28,29]. Deeper investigations of the links among imaging features and lung cancer phenotypes following the lines of a study in 2017 [29] are more rare. A promising extension to radiomics that can fill this gap is the consideration of the tumor environment, which appears in medical images as a dynamic environment that presents multiple phenotypes. This technique has been referred to as “habitat imaging” [30] and could be supported by extending radiomics from overall descriptors to voxel-based features [31].

### 2.4. Characterization/Prediction of Toxicity/Host Response to Treatment

Based on a clinical background, we know that there are some lung cancer patients who are at high risk of toxicity from radiation (interstitial lung disease, ataxia telangiectasia, scleroderma with CREST syndrome, etc.), chemotherapy (Fanconi anemia, Gilbert’s syndrome, etc.), or immunotherapy (pre-existing autoimmune disease). Understanding and predicting an individual patient’s tolerance of any given therapy is critical to understanding risk/benefit. A simple example is that of interstitial lung disease; the presence of this background condition increases the risk from stereotactic body radiotherapy twenty-fold [32]. There are likely other background diseases that have yet to be correlated with treatment toxicity, but AI could help in this regard. Future areas of research could use AI to evaluate how certain pre-existing medical conditions (or clusters of conditions) impact therapy tolerance. To our knowledge, this area remains unexplored, despite early promising results based on radiomic studies for the prediction of radiation-induced pneumonitis [33,34]. 

It might also be of interest to consider spatial RT-dose information (dosi-omics) as additional inputs to models to improve prediction accuracy. 

More generally, time-series information such as sequential images across treatment might provide additional insights than a mere analysis based on single-timepoint imaging [35]. For example, specific AI architectures such as recurrent neural networks (RNNs) can be used to encompass time-dependent information during the learning process. 

### 2.5. Integration of all Predictive/Prognostic Metrics into Summary/Composite/Ensemble “Personalized” Prediction

Ultimately, all prognostic/predictive metrics will need to be weighed against each another by prioritized optimization. In some cases, curative treatment may not be recommended because of excessive risks, whereas in others, high-intensity treatment might be deemed tolerable due to the lack of risk factors in a given patient. Cancer treatment represents a balance between cure rate and toxicity of therapy with four possibilities: (A) “nontoxic therapy, controlled cancer”; (B) “toxic therapy, controlled cancer”; (C) “nontoxic therapy, uncontrolled cancer”; or (D) “toxic therapy, uncontrolled cancer”. All patients would be happy with A and most would be happy with B, or in some cases C, but no patient would likely accept a high risk of D. This type of optimization and the resulting use of the risk estimates by both patients and clinicians will be a significant implementation challenge for any AI/ML tool, going forward.

Growing interest has been shown by commercial enterprises in the deployment of AI solutions in the clinical workflow such as auto-counting, data analytics, and automated image QA. Nevertheless, we believe that these algorithms can be improved not only by daily clinical use but also by the possibility to retrain the algorithms “on-the-fly” when, for example, manual corrections of the contours is provided by the users (“labelled data”). In this view, we foresee a commercial AI product as a dynamic entity, which can be continuously improved by its usage and is a concept similar to the user “crash/error report” when using an operating system. We are aware of the FDA pushing to consider AI applications as medical devices, with the policy to keep an updated “log-file” of wrong inference or prediction. 

## 3. AI in Medical Imaging: Dealing with Standard of Care Imaging and Confounding Factors. Are We AI Ready?

Medical images should be considered high dimensional data, embedded with more information than can be accessed via visual inspection: “images are more than pictures, they are data”. The application of AI for medical image analysis is not a new topic. In the last ten years, the number of publications defining prognostic/predictive handcrafted radiomic features or automated deep learning (DL)-based systems has increased. These publications have shown potential breakthrough applications to augment clinical decision making; however, several studies point out concerns about the reliability of such biomarkers [35,36,37]. As we indicated in some of our studies, reliability of image-derived biomarkers is strongly affected by A) an instability of prognostic values of radiomic features when validated on images acquired with different imaging/machine protocols than the ones used during model development and B) the presence of confounding factors during modelling.

The first concern is based on the fact that many radiomic biomarkers show strong dependencies on the image-acquisition settings used [37]. Medical images are acquired and reconstructed for visual inspection or semiqualitative analysis. The human eye mainly focuses on an image’s global detail; therefore, changes in the granular textures of the image will likely not impact human activities such as contouring or determination of morphological properties (e.g., size measurements) of the primary tumor. Conversely, AI systems apply multiple mathematical transformations to the original images (e.g., wavelet transforms), and many of these radiomic features are meant to measure granular textures in the images. It is not surprising that texture imaging features are the least stable with respect to changes in imaging acquisition parameters. Furthermore, many studies have shown that certain image-acquisition parameters such as smaller slice thickness (1–2 mm), inclusion of contrast medium, and or “conventional dose” instead of “low dose” imaging led to better prognostic or predictive power [37]. This evidence represents a harmonization problem when translating such biomarkers within the adaptive radiotherapy workflow. In fact, radiation delivery systems are equipped with imaging facilities, such as cone beam (CB) CT scanners, but because of hardware constraints, the quality of these images is much lower than, for example, diagnostic or planning CT scans. A recent study comparing radiomics computed on CT and CBCT clearly showed that only a small percentage of radiomic features are interchangeable between these imaging modalities [38]. A similar reasoning can be extended to images acquired within an MR-linac setup, where during treatment, the acquisition magnetic field is lower (commonly no more than 1.5T) compared to the suggested magnetic field to perform radiomic studies (3T) [39]. We are not in the position to claim that an optimal and unique image configuration for radiomics will exist because this will be dependent on the specific problem considered.

There are mitigation strategies one can consider when preprocessing medical images to increase the robustness of image-derived features: (A) correction of the dependencies of radiomic features on image acquisition settings and (B) investigation of automated methods to manipulate and improve image quality. Examples of the former have been shown in a study by Zhovannik et al., where radiomic features were corrected against changes in exposure values in CT phantom images [36], while Ligero et al. developed a post-acquisition CT image-correction method for radiomic features based on the ComBat method, which is borrowed from genomic studies [40]. A ComBat approach could work for handcrafted radiomic features but may prove challenging for neural network approaches, since ComBat bypasses, a priori, a definition of relevant imaging features to be extracted. The latter method (often referred to as domain adaptation and synthetic imaging) involves the application of advanced DL algorithms, often using generative adversarial networks (GANs). GANs are a framework that optimizes an objective by running the zero sum of a two-player game between two networks. One of the networks, called the *generator*, tries to learn the data distribution by trying to fool the other network, called the *discriminator*, which simultaneously tries to differentiate between real images and fake images created by the generator [41]. In a domain adaptation approach, the real image may be a CT scan acquired by a certain scanner manufacturer (domain, real image) that will be translated into a corresponding CT scan as acquired within another institution (target, fake image). In a synthetic imaging approach, the network could try to generate synthetic CTs from CBCTs. Figure 2a depicts an example of the latter, while Figure 2b shows how this workflow can be integrated within the current adaptive radiotherapy workflow. GANs’ applications in radiation oncology are becoming very popular and showing early promising results. For example, using GANs for domain translation improved the auto-contouring of pulmonary nodules when using imaging data from multiple institutions.

Maspero et al. trained four standard CycleGAN models on lung, breast, and head-and-neck scans—three for each anatomical site and one model for all sites. They showed that a single model for all three anatomical sites performed comparably with the models trained per anatomical site, which would simplify clinical adoption [42]. Finally, one of the major advantages of using GAN models is that the training procedure can be performed using an unpaired approach (meaning that a 1 to 1 correspondence between the images input to the network is not needed) compared to the paired approach where the network needs to receive as an input a pair of CBCT and CT of the patient acquired on the same day.

We foresee exploiting GAN models to improve the image-guided translation of radiomics in the adaptive radiotherapy workflow, with a few caveats. First, as GAN models are based on two or more networks, the computational time and complexity required to perform the training procedure is greater than with more traditional approaches. Second, synthetic images should always be verified by humans to ensure that additional artifacts or nonsense anatomical properties are not inserted into the synthetic images. The latter concern is related to false discoveries associated with high dimensional data. A recent systematic review of texture analyses of medical images pointed out that an optimal cutoff selection for tuning machine-learning predictive models leads to an increased risk of type I errors [43]. This spawned a debate about whether necessary precautions are taken into consideration when developing radiomic signatures or, in general, image-derived prognostic/prediction models. The popular paradigm that ML algorithms or more aggressive feature selection strategies can mitigate false discoveries as well as eliminate feature redundancies or confounding factors has been challenged by recent studies. For example, Welch et al. showed that radiomic features are susceptible to underlying dependencies and multicollinearity within models [44]. Therefore, radiomic models and features must be tested to determine added prognostic and predictive accuracy compared to accepted clinical factors. In our recent study, we showed how ML can be used at the early stages of model building to evaluate and reduce the presence of confounding factors [45]. ML unsupervised methods, such as kernel principal component analysis (PCA) or hierarchical clustering, can help identify intercorrelations as well as dependencies between imaging features and clinically accepted prognostic factors (e.g., tumor extension), as we have successfully shown in CT scans from lung cancer patients [45].

In summary, we have shown how AI, more specifically ML and DL algorithms, can be used in the preliminary stages of the model-building computational chain in order to augment image quality and to improve the methodological aspects connected to the development of radiomic signatures.

While we focused on describing a methodological workflow for radiomics, a similar concept holds for deep learning. If on one side the complexity of deep neural networks can measure high-level imaging features less dependent on image acquisition settings, on the other, deep learning models still require extensive external validation as well as meticulous effort in improving the image quality of input data. 

## 4. AI for Biomarker Discovery. from Medical Imaging to Biology

Biological aspects of tumors are known to greatly influence treatment response. For example, hypoxia, a common feature of solid tumors, is a strong prognostic factor and is known to limit the efficacy of standard-of-care chemotherapy and radiotherapy [46]. On the other hand, molecular alterations in the tumor also provide new targets for anticancer treatment. Treatment agents targeting multiple cellular pathways involved in tumorigenesis (i.e., EGFR) [47] and immunotherapy have shown efficacy in tumors with a high mutational border and high PDL1 expression [48]. Accurately mapping tumor biology is essential in selecting the most optimal, personalized treatment regimen for a patient. AI can aid clinicians by using medical imaging to noninvasively collect imaging-derived biomarkers that predict intrinsic tumor biology.

Several factors play roles in the radio- and chemoresistance of solid tumors, including hypoxia, DNA-repair deficiencies, and cellular senescence [49]. Among these features, hypoxia offers the possibility to be analyzed using advanced imaging techniques such as specific hypoxia positron emission tomography (PET) tracers [50] and diffusion-weighted MRI [31]. In NSCLC, the noninvasive detection of hypoxia has been demonstrated for a combination of PET/CT and dynamic contrast-enhanced CT [51]. AI has been employed to develop disease-specific radiomic hypoxia classification signatures [52]. A radiomic signature comprised of four CT-derived features was identified for lung cancer that reached an area under the curve (AUC) of 0.80 (95%CI 0.65–0.95) for the prediction of tumor oxygenation status in an external validation cohort [53]. Radiomic features derived from hypoxia-tracer PET imaging have also been used to develop multivariate prognostic models in malignant glioma (5% relative risk prediction performance increase for overall survival) [54] and hypoxia-based patient stratification in head-and-neck cancer [55]. All in all, noninvasive prediction of oxygenation status can aid in the prediction of radio- and chemoresistance as well as patient stratification for hypoxia-targeting therapies. Additionally, a ML-based model has been developed to predict DNA mismatch repair deficiency, a contributing factor in radio- and chemo-resistance for endometrial cancer. This model reached an AUC of 0.78 (95%CI 0.58–0.91) [56]. These examples show the potential of radiomic-based models in predicting key tumor biological factors implicated in radio- and chemoresistance.

In NSCLC, several cellular pathways in which targetable mutations occur have been identified. Alterations in two of these pathways, including mutations in EGFR and ALK gene rearrangements, have led to the inclusion of tyrosine kinase inhibitors in the standard-of-care treatment for this subset of metastatic NSCLC patients (and, more recently, for EGFR-mutated LA-NSCLC patients post-surgical resection) [57]. Several other pathway inhibitors are currently being studied in clinical trials [4]. AI approaches can aid in treatment decisions by identifying the presence of molecular alterations and by predicting treatment response or acquired resistance to treatment. Several attempts have been made to use radiomics signatures to predict EGFR status and other molecular targets in NSCLC patients [58]. For example, a ML model using CT radiomics and clinical features achieved a diagnostic accuracy of 88.3% in the external validation dataset for predicting EGFR mutant NSCLC [59]. Additionally, PET-imaging derived radiomic features have also been used to predict EGFR mutation status with accuracies around 75–78% [60,61]. The development of the T790M mutation in EGFR, which can occur during treatment with first-generation EGFR tyrosine kinase inhibitors (gefitinib and erlotinib) is an important mechanism of resistance. Radiomics signatures have also been developed to predict the development of this mutation [59].

An important clinical challenge is the observation that not all patients who harbor a specific molecular alteration (EGFR mutation) respond to tyrosine kinase inhibitors. CT-based ML models were developed to identify stage IV EGFR-mutated NSCLC patients who are not likely to benefit from EGFR-targeted therapy (HR 2.13, 95%CI 1.30–3.49) [62]. Similarly, a radiomic signature was identified that could significantly risk-stratify ALK-positive NSCLC patients (HR 2.181, *p* < 0.001) when treated with ALK-inhibitor critzotinib [63]. This could aid clinicians in more optimal patient stratification for tyrosine kinase inhibitors.

A key development in the treatment of NSCLC is the approval of the immune checkpoint programmed death 1 (PD1) inhibitor durvalumab in the standard-of-care treatment of LA-NSCLC [64]. The expression levels of PDL1 as well as the tumor mutational burden (TMB) have been proposed as two biomarkers predictive for the response to immune checkpoint inhibitors.

A radiomic biomarker based upon CT-derived deep learning features was able to distinguish between high- and low-TMB groups (AUC 0.81, 95%CI 0.77–0.85) in the test cohort [65]. Similarly, radiomic models were developed to predict PDL1 expression levels (>1% positivity) with high accuracy for CT features (AUC 0.97 (95%CI 0.93–1.0) [66] and to predict PDL1 expression (>50% positivity) by using CT radiomic features combined with clinicopathological features (AUC 0.848) [67].

Furthermore, AI models have been tested to predict responses to immune checkpoint blockers. A radiomic biomarker was able to stratify patients treated with anti-PD1/PDL1 immunotherapy into two risk cohorts (HR 0.54, 95%CI 0.31–0.95) [66]. FDG-PET radiomics was also able to predict survival in patients with >1% expression levels of PDL1 receiving pembrolizumab with 78% (SD 18%) accuracy. Similar results for PET-based radiomics have also been reported in other studies with AUC values varying from 0.80 to 0.86 [68,69].

Another phenomenon observed in NSCLC patients treated with immune checkpoint inhibitors is a paradoxical acceleration of tumor growth (hyper progression) after the initiation of treatment. Radiomic features from pretreatment CT scans were able to stratify patients at risk for this hyper progression with an AUC of 0.96 in the validation set [70]. Together, these examples present the opportunities for employing AI in predicting the response to immunotherapy and aid in the accurate stratification of patients.

## 5. AI for ART Workflow Optimization

It is recognized that the adaptation of radiotherapy could be disruptive to the radiotherapy workflow and may require additional time and resources, which would further tax an already complex and time-critical process. Hence, advanced analytics such as AI are not only useful as efficient and time-saving tools but also a necessary requirement to meet the demands of such a process. The two key elements are automation and optimized decision-making.

Automation has been the subject of intense research in RT and traces its roots to the utilization of onboard imaging techniques such CBCT (and, more recently, MRI scans) for treatment setup. AI methods can accelerate the replanning processes in IGART via contour propagation, image registration, and dose recalculation [71]. As previously mentioned, synthetic image generation and domain adaptation (for example, from MR to synthetic CT) may become a necessary part of treatment adaptation and the online replanning workflow for hybrid treatment devices such as an MR-linac. However, a more intriguing and rather challenging aspect of ART is reoptimizing the required dosage to improve clinical outcomes. This goes beyond accounting for geometrical changes into improving the therapeutic ratio of better tumor control to less side effects. This has been an active area in radiotherapy research employing divergent techniques such as generalized TCP/NTCP models [72] as well as the use of advanced AI and deep learning [73], as we successfully showed. For instance, Tseng et al., proposed an ART system to estimate the adaptive dose per fraction using a 3-component deep reinforcement learning (DRL) approach with a neural network architecture (Figure 3). The DRL architecture was composed of (1) a GAN to learn patient population characteristics to overcome training from a limited sample size, (2) a radiotherapy artificial environment (RAE) reconstructed by a deep neural network (DNN) utilizing both original and synthetic data (by GAN) to estimate the transition probabilities for adaptation, and (3) a deep Q-network (DQN) applied to the RAE for choosing the optimal dose in a response-adapted treatment setting. Interestingly, the DRL seemed to suggest better decisions than the clinical ones in terms of mitigating toxicity risks and of improving local control, as seen in Figure 3.

## 6. Deployment of Decision Support Systems

The availability of vast quantities of electronic clinical data and the possibility of real-time streaming from monitoring and treating devices coupled with the falling cost of high-performance computers in the clinical setting has led to a rush of developments in intelligent clinical decision support systems (iCDSS). An iCDSS is defined as an active knowledge software system that incorporates two more or patient data points to generate situation-specific advice that aids physicians with clinical decision-making. “Active knowledge” relates to algorithms, such as predictive or prognostic models, housed inside the iCDSS, that can learn from data without being hard coded to provide the expected response.

Progress in the fields of molecular biology, genomics, proteomics, and quantitative image analysis has led to a wide range of “omics” that can be mined for prognostic and predictive biomarkers. With the aforementioned explosion in available information and the number of treatment options, this makes it more difficult for physicians to make evidence-based choices. AI can be given the task of filtering, extracting essential signals from data, and then presenting this condensed information in a way that it is more cognitively accessible to the human mind. However, despite thousands of potentially clinically relevant AI-based models being published in recent years, few actually make it into an iCDSS that is regularly used in the clinic.

One of the most commonly quoted barriers is the perception that many AI-based decision support systems are “black box” applications. That is, the internal logic of some of these systems may be very difficult to deduce from “outside of the box”, since the only apparent way to infer the internal logic seems to be to observe the output for a wide range of test inputs. In the radiotherapy context, some analogies have been drawn between specification, acceptance testing, and commissioning of iCDSS with other radiotherapy devices, such as treatment planning software and radiation generators. Improving the interpretability of AI models is an active area of current research, producing helpful tools such as activation and attention maps, and Shapley additive explanations (SHAP) values [74,75]. However, depending on the methods and features used, even such interpretability tools may not be furnished much by way of rationalizing a human-like clinical decision.

The “black box” argument arises because of concern about algorithm bias and potential discrimination entrained in the model. Bias could arise because of unaccounted differences in practice context, patient characteristics, or imaging settings or be a direct result of ingrained discrimination and prejudices within human society that are imprinted into the training data. Patients can be given legal protection to demand a meaningful explanation of the logic used to reach a decision and have recourse to challenge such a decision in court [76]. Extensive testing in different practice settings and independent external validation are key to intercepting algorithmic bias, thus increasing clinical confidence.

The second barrier to adoption of iCCDS is the lack of emphasis on independent external statistical validation. Focus on model development alone is inevitably misplaced because, without strong evidence of generalizability, there would be little confidence that an iCCDS works as required in a different clinic, region, country, or even similar population but at a different time.

Assuming an iCCDS is finally deployed into clinical practice, it is important to distinguish reactive versus proactive approaches for ensuring fitness for clinical purpose. The former is analogous to quality control (QC) in radiotherapy, where one implements measures to intercept non-conforming performance by the iCCDS. The inevitability of concept drift, where the implicit relationship between the input variables and the desired outcome shifts (for example, due to technological evolution or introduction of new classes of drugs), suggests that the latter approach may be more appealing. Rather than correcting an integrated system only after it fails a predefined set of criteria, a continuous evaluation paradigm seeks to detect drift with statistical procedures similar to industrial process control. This allows for the option of preventative maintenance before system performance falls outside of operational criteria.

Figure 4 illustrates the alignment of iCCDS development with continuous evaluation to a development-operational (DevOps) software lifecycle philosophy. Rather than viewing the creation of an iCCDS as the conclusion to a process (right half of the cycle), development needs to merge seamlessly into extensive testing in the real world, with commissioning and routine quality assurance (left half of the cycle). Understanding the clinical need is paramount, since an iCCDS must exist to address a clear and present requirement and the continuous evaluation must ensure that an operational iCCDS always remains fit for its purpose to serve that clinical need.

## 7. Conclusions

In this review, we present a comprehensive roadmap for using AI to support improved management of LA-NSCLC patients. This roadmap starts with the correct identification of clinical unmet needs and clinical research questions, which represent challenges in clinical decision-making for treating physicians. Only after the correct identification of these clinical questions should meaningful AI applications be developed to support the release of clinical decision support systems. We provide an overview of the other potential milestones where AI applications can be crucial. Our roadmap integrates multi-disciplinary inputs from the radiation oncology and computational domains and provides a new perspective on developing AI for lung cancer patients as part of a communitarian effort.

## Figures and Tables

**Figure 1 cancers-13-02382-f001:**
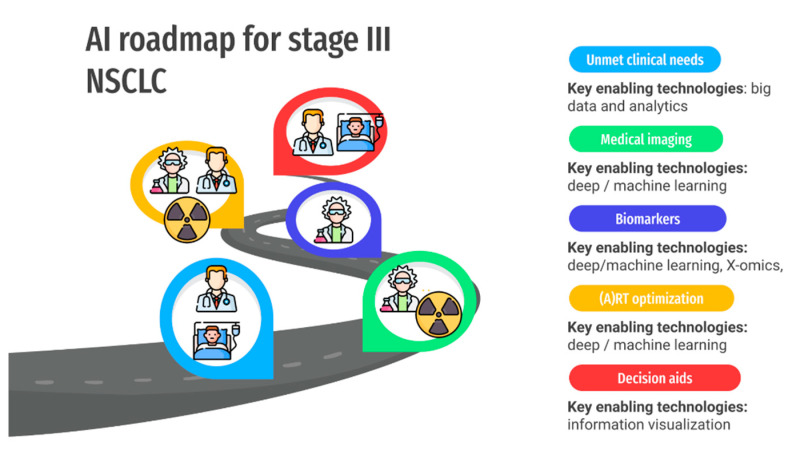
An overview of the critical points and key enabling technologies for the application of AI technologies to improve the current IGART workflow in the management of LA-NSCLC patients. Each of the critical points includes different stakeholders who need to be involved for a productive and collaborative environment. The structure of this review follows the progression of this figure.

**Figure 2 cancers-13-02382-f002:**
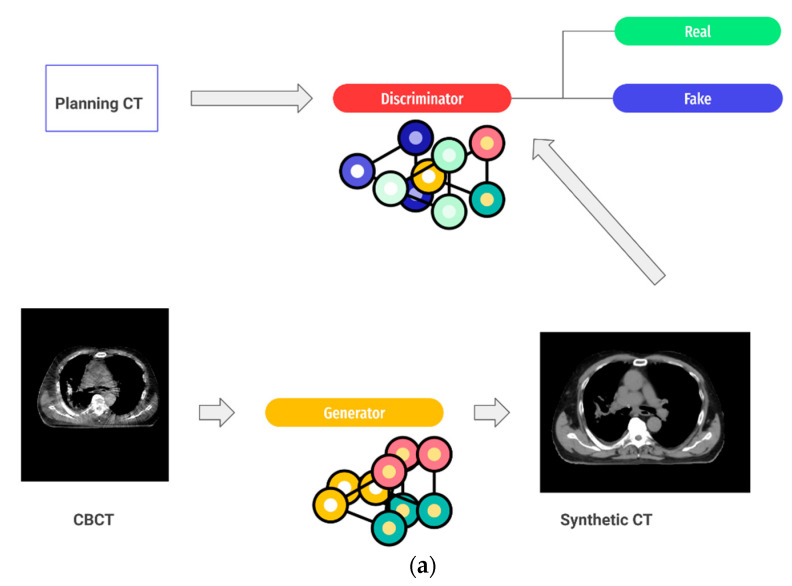
(**a**) GAN framework showing CBCT to CT image translation. The generator tries to learn the data distribution of CBCT and CT images and uses this learned representation to convert a CBCT image into a “fake” CT image. The generator learns this representation by trying to fool the discriminator while it compares real and “fake” CTs and learns to differentiate between them. (**b**) Workflow for adaptive radiotherapy enabled by dose calculation from CBCT images. Prior to dose recalculation, a CBCT image is converted to a synthetic CT (sCT), providing correct Hounsfield unit (HU) values and removing artifacts while preserving the anatomy.

**Figure 3 cancers-13-02382-f003:**
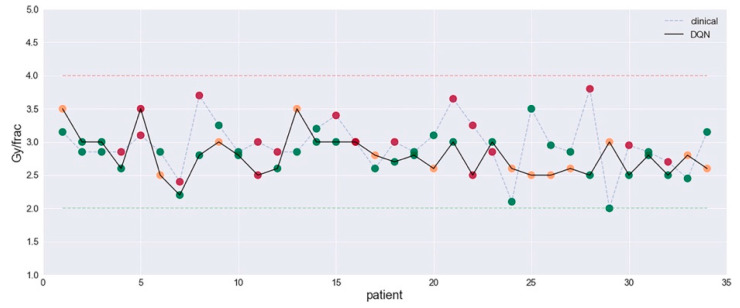
A response-based ART system using a three-component deep reinforcement learning (DRL). Automated dose decisions given by DQN (green dots) vs. clinical decision (blue dots) with RMSE 0.76 Gy. Dots in green and red represent indications of “good” and “bad”. Reproduced with permission from Tseng, Med Phys, 2017.

**Figure 4 cancers-13-02382-f004:**
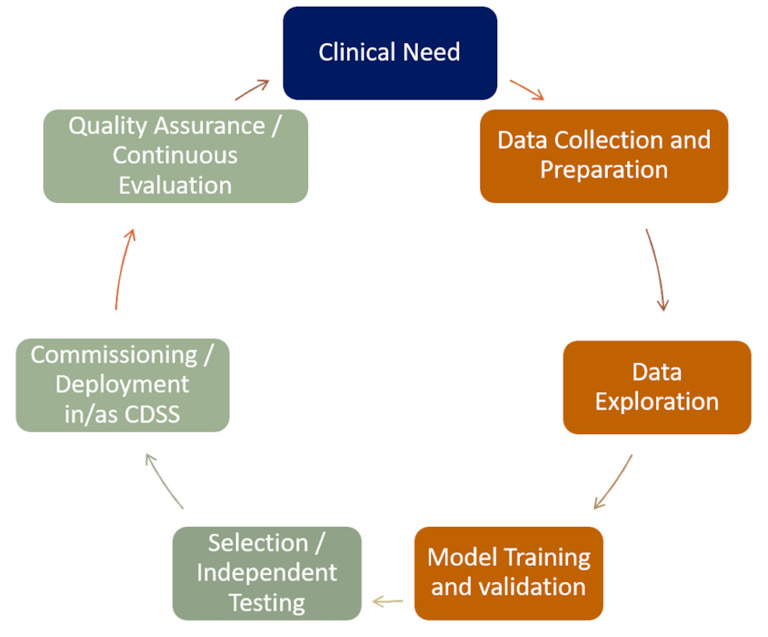
Overview of the interconnections among the developments of an iCCDS and the traditional software lifecycle schema.

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
