# Peer review of "Artificial Intelligence Applications to Improve the Treatment of Locally Advanced Non-Small Cell Lung Cancers"

_cancers, 2021, doi:10.3390/cancers13102382_

Round 1

Reviewer 1 Report

This review paper is well written and I only have minor comment, which might help further improve the quality of this manuscript. This research addressed various AI applications for locally advanced non-small cell lung cancers. This is a review paper, which covers the topic of medical imaging, adaptive RT and biomarkers. This review paper covers some important topic and challenge in clinical locally advanced non-small cell lung cancer.

  1. line 338, Figure 2,  the synthetic CT image is not real synthetic CT image and it might be better to replace the current "red" image with real synthetic image.
  2. Line 353, 1:1 -> 1 to 1 corresponding

conclusions are consistent with the evidence and arguments presented and they address the main question posed.

Reviewer 2 Report

The authors have written a comprehensive overview of the literature regarding AI application in the radiotherapy workflow with a special focus on locally advanced NSCLC. The literature review is overall well-balanced, nicely written and produces a well structured path of thought to follow. However, there are some minor remarks that I would advice to consider before publication of this work:

1)Them term “Radiomics” is not being properly introduced until section 3). The authors should move the description of the Radiomics principle into in the introduction as they already use the term in setion 2.

2) Section 2.4. Toxicitiy prediction

A few “exploratory” analyses exist from the early time of radiomics studies that could be cited as potential ways of application:

Krafft SP, Rao A, Stingo F, et al. The utility of quantitative CT radiomics features for improved prediction of radiation pneumonitis. Med Phys [Internet]. 2018; 45: 5317–24. Available at: https://aapm.onlinelibrary.wiley.com/doi/abs/10.1002/mp.13150?af=R

Cunliffe A, Armato SG, Castillo R, et al. Lung texture in serial thoracic computed tomography scans: correlation of radiomics-based features with radiation therapy dose and radiation pneumonitis development. Int J Radiat Oncol Biol Phys [Internet]. 2015 [cited 18 October 2016]; 91: 1048–56. Available at: http://www.ncbi.nlm.nih.gov/pubmed/25670540

3) Section 2.4. Toxicitiy prediction

Using the spatial RT-dose information as an additional input for future toxicity prediction models may be another idea to mention in this paragraph (“Dosiomics”).

4)Section 3: The authors nicely summerarize the problem of reproducibility of Radiomics. However, the cited studies predominantly focus on “hand-crafted” features. In my opininon, external validation of deep learning models is less frequent in the current literature. The authors should provide also evidence on Deep learning models and their dependency on aqusition protocols, scanner types etc. Are they potentially less dependent on acqusition parameters?

5) The authors describe scientific studies in areas of AI applications that have been introduced into the clinic.

Section 2: OAR deep learning based contouring. Multiple software solutions are available now.

Section 3: Replanning based on GAN-derived syngthetic CTs from CBCTs. A commerical product allowing adaptive RT recently entered the market (Ethos system by Varian).

In both cases, it would be interesting for the reader how these current solutions already represent the mentioned applications of AI and what additional developments are necessary to leverage the full suspected potential.

6) One additional point that the authors could discuss is the possibility to analyse longitudinal data. For instance, Xu et al demonstrated improved prognostic assessment after integration of follow-up CT images into a Deep Learning workflow. This concept could also be applied in other areas of AI applications, as well.

Xu Y, Hosny A, Zeleznik R, et al. Deep Learning Predicts Lung Cancer Treatment Response from Serial Medical Imaging. Clin Cancer Res [Internet]. 2019; 25: 3266–75. Available at: http://clincancerres.aacrjournals.org/lookup/doi/10.1158/1078-0432.CCR-18-2495

Author Response

Please see attatchment.
